# Alteration of Fatty Acid Profile in Fragile X Syndrome

**DOI:** 10.3390/ijms231810815

**Published:** 2022-09-16

**Authors:** Armita Abolghasemi, Maria Paulina Carullo, Ester Cisneros Aguilera, Asma Laroui, Rosalie Plantefeve, Daniela Rojas, Serine Benachenhou, María Victoria Ramírez, Mélodie Proteau-Lemieux, Jean-François Lepage, François Corbin, Mélanie Plourde, Mauricio Farez, Patricia Cogram, Artuela Çaku

**Affiliations:** 1Centre de Recherche du CHUS, Department of Biochemistry, Faculty of Medicine and Health Sciences, Université de Sherbrooke, Sherbrooke, QC J1H 5N4, Canada; 2Department of Child Neurology, Raúl Carrea Institute for Neurological Research (FLENI), Buenos Aires C1428AQK, Argentina; 3Centre de Recherche sur le Vieillissement, Departments of Medicine, University of Sherbrooke, Sherbrooke, QC J1H 4C4, Canada; 4Department of Pediatrics and Centre de Recherche du CHUS, University of Sherbrooke, Sherbrooke, QC J1H 5N4, Canada; 5Biomedicine Division, Centre for Systems Biotechnology, Fraunhofer Chile Research Foundation, Santiago 7500588, Chile

**Keywords:** fatty acids, Fragile X, phospholipids, lipids, neurodevelopmental disorder

## Abstract

Fragile X Syndrome (FXS) is the most prevalent monogenic cause of Autism Spectrum Disorders (ASDs). Despite a common genetic etiology, the affected individuals display heterogenous metabolic abnormalities including hypocholesterolemia. Although changes in the metabolism of fatty acids (FAs) have been reported in various neuropsychiatric disorders, it has not been explored in humans with FXS. In this study, we investigated the FA profiles of two different groups: (1) an Argentinian group, including FXS individuals and age- and sex-matched controls, and (2) a French-Canadian group, including FXS individuals and their age- and sex-matched controls. Since phospholipid FAs are an indicator of medium-term diet and endogenous metabolism, we quantified the FA profile in plasma phospholipids using gas chromatography. Our results showed significantly lower levels in various plasma FAs including saturated, monosaturated, ω-6 polyunsaturated, and ω-3 polyunsaturated FAs in FXS individuals compared to the controls. A decrease in the EPA/ALA (eicosapentaenoic acid/alpha linoleic acid) ratio and an increase in the DPA/EPA (docosapentaenoic acid/eicosapentaenoic acid) ratio suggest an alteration associated with desaturase and elongase activity, respectively. We conclude that FXS individuals present an abnormal profile of FAs, specifically FAs belonging to the ω-3 family, that might open new avenues of treatment to improve core symptoms of the disorder.

## 1. Introduction

Fragile X Syndrome (FXS) is an X-linked neurodevelopmental disorder known to be the most prevalent hereditary monogenic cause of Intellectual Disability (ID) and Autism Spectrum Disorders (ASDs) [1,2]. FXS is caused by a mutation of the *Fragile X Messenger Ribonucleoprotein 1* (*FMR1)* gene (*FMR1 Fragile X Messenger Ribonucleoprotein 1 [Homo Sapiens (Human)]—Gene—NCBI*, n.d.), leading to a deficit in the Fragile X Mental Retardation Protein (FMRP) [3]. It is well established that the decrease in expression of FMR1 affects the brain and cognitive functions [4]. However, the ubiquitous expression of FMRP in the body suggests a specific role in other organs [5]. Specifically, preclinical and clinical studies have shown that the lack of FMRP is associated with metabolic alterations, including a decrease in body fat storage as well as a reduction in plasma glucose, glycerol, and cholesterol levels [6,7,8,9].

Lipid content represents more than half of the brain’s dry weight, and fatty acids are among the most crucial lipid molecules that determine the brain’s integrity and function [10]. Fatty acids (FAs) classify into three categories: saturated FAs (SFAs, with no double bonds), monounsaturated FAs (MUFAs, with a single double bond), and polyunsaturated FAs (PUFAs, with ≥2 double bonds). Further, PUFAs, due to the position of their first double bond on methyl terminals, are classified into omega-3 (ω-3) and omega-6 (ω-6) FAs [11]. PUFAs include important structural and functional FAs such as the essential FAs of linolenic acid (ALA, 18:3 ω-3) and linoleic acid (LA, 18:2 ω-6), which are metabolized by elongation and desaturation into eicosapentaenoic acid (EPA, 20:5 ω-3), docosahexaenoic acid (DHA, 22:6 ω-*3*), and arachidonic acid (AA, 20:4 ω-6) via the elongase and desaturases enzymes [12,13]. Adequate FA intake is important for brain development, perfusion, inflammatory processes, and the synthesis of neuroprotective metabolites [12,14]. Various studies have shown that the lack of ALA has a behavioral impact on both animal and human models.

Changes in the FA profile of various tissues, including plasma, red blood cells, and post-mortem brain, have been reported in various neuropsychiatric disorders such as attention deficit hyperactivity disorder (ADHD), schizophrenia, anxiety disorders, ID, and ASD [15,16,17,18,19]. Specifically, low levels of plasma ALA, LA, EPA, DHA, and AA have been reported in children with autism compared to the controls [20,21]. Moreover, supplementation with FAs, particularly ω-3 FAs, including ALA, EPA, and DHA, has shown improvements in autistic behavioral symptoms [22,23,24]. Despite these findings in autism, a disorder for which FXS is the prototypic model, to the best of our knowledge, the FA profile in FXS individuals has not been studied. The present work aimed to characterize the FA profile of plasma phospholipids in FXS individuals as compared to healthy controls for the first time. Since FXS is the most prevalent hereditary monogenic cause of intellectual ID and ASD, we hypothesized that FXS individuals would show lower levels of plasma FAs compared to controls, which may provide new insights into treatment strategies for FXS.

## 2. Results

### 2.1. Study Population Characteristics

The study population included two groups comprising two ethnic cohorts, as shown in Table 1. The majority of the participants were male. All Argentinian FXS individuals had the full FMR1 mutation, while 11% of the French-Canadian cohort were mosaics. There was no statistical difference in age and sex between FXS individuals and controls. The fasting conditions applied to Argentinian and French-Canadian cohorts were 6 and 12 h, respectively.

### 2.2. FA Profile

FXS participants showed lower levels for all FAs as compared to controls. The statistical significance was reached for the majority of SFAs, MUFAs, and PUFAs, as shown in Table 2. Specifically, significant lower levels were shown in the total amount of MUFA and PUFA families, with more significancy in PUFA (*p* = 0.0081 and *p* = 0.0015, respectively), including both ω-3 and ω-6 with, respectively, 22% and 16% lower levels of FAs in FXS compare to controls. Comparable results were obtained when male FXS were compared to male controls (Appendix A)

Although our results from age-related analysis showed significant differences in FA levels of FXS adults compared to controls (Appendix A), interestingly children and adolescent cohorts did not show any difference in the level of FAs compared to controls (Appendix A, respectively).

#### 2.2.1. ω-6 FA Profile

All ω-6 FAs were significantly lower in the FXS group compared to the matched controls (Table 2 and Figure 1). LA (C18:2 ω-6) was significantly lower in both cohorts, Argentinians (14%, median in FXS, 389.4 mg/L, vs. controls, 451.7 mg/L, *p* = 0.0246) and French-Canadians (39%, median in FXS, 307.6 mg/L, vs. controls, 400.6 mg/L, *p* = 0.0030) compared to the controls, as shown in Appendix A.

#### 2.2.2. ω-3 FA Profile

There was a decreasing trend for all four measured ω-3 FAs, including ALA (C18:3 ω-3), EPA (C20:5 ω-3), DPA (C22:5 ω-3), and DHA (C22:6 ω-3), as shown in Table 1 and Figure 2. The EPA was lower in both the FXS cohorts of Argentinians (31%, median in FXS, 4.8 mg/L, vs. controls, 9.8 mg/L) and French-Canadians (50%, median in FXS, 10.2 mg/L, vs. controls, 14.5 mg/L) compared to the controls, as shown in Appendix A. Moreover, the DHA was lower in both FXS cohorts, Argentinians (19%, median in FXS, 47.6 mg/L, vs. controls, 58.1 mg/L) and French-Canadians (51%, median in FXS, 39.5 mg/L, vs. controls, 80.8 mg/L) compared to the controls, as shown in Appendix A.

### 2.3. Estimation of Metabolical Enzymes Activity

The ratio of specific FAs was used to estimate the activity of enzymes involved in FA metabolism. There was a significant decrease in the EPA/ALA (20:5/18:3 ω-3; estimation of delta5 and delta6 desaturase activity) ratio in the FXS group as compared to the controls (medians, respectively, 2.4 vs. 3.1, *p* = 0.0410). Likewise, there was an increase in the DPA/EPA (22:5/20:5 ω-3; estimation of elongase activity) ratio in the FXS group as compared to the controls (medians, respectively, 2.5 vs. 1.6, *p* = 0.0455). There was no difference in the ratios of either the 20:4/20:3 or 20:3/18:3 ω-6 FA biosynthesis pathways.

## 3. Discussion

The metabolic basis of FXS pathology and its impacts on the clinical phenotype are poorly understood. In this study, we performed FA profiling in two distinct cohorts to explore if alterations of FAs reported in ASD are similarly found in FXS. To the best of our knowledge, this report is the first to measure FAs content in plasma phospholipids in humans with FXS as compared to age and sex matched controls.

Our results show a decreasing trend of all measured FAs in FXS as compared to controls, reaching the statistical significance for the majority of SFA, MUFAs, and PUFAs. FAs are involved in the biosynthesis of different lipid structures including sterols, phospholipids, eicosanoids, and glycolipids [25]. We have previously reported lower plasma cholesterol levels in FXS individuals as compared to healthy controls [7]. Taken together, our findings suggest a reduction in all plasma lipids in FXS. Since both FAs and cholesterol are the most abundant lipids in the human brain [26] and are essential for cell membrane integrity and synaptic development [27,28], a reduction in plasma lipid content might have an impact on membrane cell structure and stability. We recently showed that FXS individuals display a reduction in cholesterol content in platelet-derived lipid rafts that were associated with specific traits of aberrant behavior [29]. Indeed, lipid rafts are dynamic structures enriched in lipids that recruit important synaptic receptors (mGluRs, NMDAR, and AMPAR) and facilitate protein–protein and protein-lipid interactions of intracellular signaling molecules [30]. Since platelets are used as a proxy model of human neurons, further lipidomic studies are warranted to explore if FAs are decreased as well. We thus hypothesize that the decrease in total lipids in FXS might have a mechanistic implication in FXS pathophysiology through an alteration of lipid raft content. Specifically, our results showed a significantly lower level of PUFAs, including EPA and DHA in FXS individuals as compared to controls. Consistent with our results, a study on 153 autistic children showed a marked reduction in DHA (22:6 ω-3) levels in plasma phospholipids [31]. Another study on plasma phospholipid fatty acids of 15 ASD children showed lower levels of DHA as compared to 18 children with Intellectual Disability, which resulted in lower levels of total ω-3 and, consequently, a 25% decrease in the ω-3/ω-6 ratio [32]. Comparable results were also reported in ADHD, another disorder for which the phenotype partly overlaps with that of FXS. Specifically, the phospholipid analysis of erythrocytes showed significantly lower levels of DHA and total ω-3; a higher level of total ω-6 and a lower ratio of ω-3/ω-6 FAs in 11 ADHD patients compared to 12 healthy controls [33]. Considering the high prevalence of ASD and ADHD in FXS (respectively, up to 50% and 59%), the consistent findings of FA profile alterations suggest that ω-3 FAs are potentially involved in FXS pathophysiology [34,35]. Indeed, preclinical studies in FXS showed that the supplementation of ω-3 FAs rescued some aspects of Fragile X phenotypes in *fmr1*-KO mice and rats [36,37]. Specifically, the ω-3 FA supplementation of *fmr1*-KO mice with a PUFA-enriched diet, containing 11.2% of EPA and 6.5% of DHA, improved their behavioral abnormalities, including alterations in emotionality, social interaction, and non-spatial memory [36]. Moreover, the impact of ω-3 PUFA dietary supplementation was recently investigated on the altered behavior of *Fmr1-^Δ^exon 8* rats, another FXS animal model associated with FXS [37]. Their deficits in the social and cognitive domains were counteracted by perinatal ω-3 PUFA supplementation. To the best of our knowledge, there has been no clinical study of FA supplementation on FXS humans, while clinical trials in ASD and ADHD are numerous and advocate that ω-3 supplementation might improve their behavioral phenotype. For instance, early studies in ASD showed that ω-3 FA supplementation with 0.93 g of EPA and DHA for 6 weeks [38] or with 135 mg EPA and 90 mg DHA for 3 months [39] had therapeutic effects on the language and maladaptive behaviors in ASD children. Another open-label study of EPA and DHA supplementation (190 mg of EPA acid and 90 mg of DHA per 500 mg capsule; 2 capsules per day) on ten ASD children showed an improvement on the Autism Treatment Evaluation Checklist [40]. However, a recent double-blind, randomized placebo-controlled study on 54 ASD children showed that supplementation for 6 months with ω-3 FAs (including 800 mg/day of DHA and 25 mg/day of EPA) had no impact on clinical test scores [41].

Meanwhile, a double-blind randomized placebo-controlled trial with 40 boys with ADHD and 39 matched healthy controls showed that daily ω-3 supplementation with 650 mg of EPA and 650 mg of DHA reduced ADHD symptoms after 4 months of treatment [42]. Another double-blind placebo-controlled study of ω-3 FA supplementation with 600 mg EPA and 120 mg DHA for 16 months improved the working memory function in 55 ADHD children [43]. The studies on ASD and ADHD suggest that various ratios of EPA to DHA in ω-3 FA supplements have inconsistent effects on the clinical profile [41,43,44]. DHA and EPA are structural components of cell membranes in the central nervous system. DHA contributes to neuronal growth and differentiation, while EPA has important anti-inflammatory functions [45,46]. Considering the important roles of both these FAs as well as the fact that the majority content of EPA can be converted into DHA [47], we hypothesize that a supplementation formula with a higher EPA content than DHA might be more effective in the treatment of neurodevelopmental disorders, including FXS.

In the current study, we also reported a significant decrease in LA for both cohorts. ALA showed a decreasing trend in FXS individuals as compared to healthy controls, although it did not reach statistical significance. In parallel to our result, two studies in ASD children showed a significant decrease in free plasma LA and ALA as compared to healthy controls [31,48]. The conversion of LA and ALA into their relevant PUFA products is mediated by elongases and desaturases. Specifically, delta5-desaturase and delta6-desaturase are key enzymes of PUFA desaturation [49,50,51]. The two enzymes are very competitive between the ω-6 and ω-3 families, including LA and ALA, as well as their downstream PUFA metabolites such as AA, EPA, and DHA [52]. In the present study, we observed a significant decrease in the EPA/ALA ratio and a significant increase in the DPA/EPA ratio in FXS patients as compared to controls. No significant differences were seen for the ω-6 ratios, suggesting that both desaturases and elongase might be more engaged with ω-6 than ω-3 metabolism. Considering that dietary FA intake along with desaturases determines PUFA quality and quantity [53], further research including food questionnaires and an RNA expression analysis are needed to explore a potential alteration of these enzymes’ activity.

The results of the current study should be considered in light of some limitations. The effects of several factors on the FA profile such as sex, diet, and BMI were not monitored in this study. Since FXS is highly prevalent in males, the limited number of recruited females did not allow us to draw female-related conclusions. Considering difficulties in overseeing the adequate intake of ω-6 and ω-3 through food questionnaires [54], it is not clear if dietary records would have been relevant in the FXS population. In the present study, we quantified FAs in plasma phospholipids, which reflect the dietary intakes during the past few weeks and are not influenced by recent FA intake [55,56]. Moreover, our subgroup analysis showed a significant decrease in DHA and EPA only in adults with FXS. However, the small sample size of children and adolescent participants provides inconclusive results for these age groups. Further studies including young children are warranted to characterize the FA profile abnormalities and, thus, to determine if there is an age range for early intervention in terms of FA supplementation that could benefit this population. In addition, we did not include any data about the clinical profile. For both cohorts, the available clinical data including behavioral questionnaires were inconsistent and insufficient to perform a correlation analysis. Considering the positive results of preclinical studies in FXS and clinical studies in the ASD of PUFA supplementation, we support the inclusion of phenotype evaluations for future studies targeting FA profiles. The identification of clinical traits related to specific PUFAs will help identify potential outcomes for future trials with FA supplementation. Since FXS is a rare disease, it is difficult to recruit and reach a large sample size, although multi-centric studies can help to perform larger scale population studies to validate our results.

In conclusion, the consistent findings in both the Argentinian and Canadian cohorts suggest a perturbation of the FA profile of FXS individuals, in particular of various FAs belonging to the ω-3 family. These observations are somehow consistent with the previous studies on other neurodevelopmental disorders including ASD or ADHD. Our results provide new directions for preclinical research to explore the underlying causes and effects of low plasma lipids in FXS and further explore treatment strategies, including supplementation with EPA and/or DHA.

## 4. Materials and Methods

### 4.1. Study Population

The study population included two groups: (1) the FXS group including 23 Argentinian FXS patients and 11 French-Canadians FXS individuals recruited at the Neurology Department of FLENI hospital (Buenos Aires, Argentina) and the Fragile X Clinic of the CIUSS de l’Estrie-CHUS (Sherbrooke, QC, Canada), respectively; (2) the control group including age- and sex-matched healthy individuals, recruited through Neurology Department of FLENI hospital (Buenos Aires, Argentina) and the general population of Sherbrooke (Quebec, Canada).

For the Argentinian FXS cohort, the inclusion criteria comprised (1) males and females older than 4 years of age; (2) a diagnosis of Fragile X Syndrome with molecular genetic confirmation; (3) the ability to undergo a blood draw without sedation or anesthesia; (4) an authorized legal representative (usually father, mother or guardian) having a level of comprehension that allows them to understand the study and its procedures and to thus sign an informed consent form; and (5) permitted concomitant medication being stabilized for at least 4 weeks prior to the blood draw. We excluded the individuals who (1) were treated with dopaminergic drugs, N-methyl-D-aspartate (NMDA) antagonists, tricyclic antidepressants, typical antipsychotics, and/or lithium; (2) have been treated with an experimental drug within the four weeks prior to signing the informed consent form or who are receiving treatment with an experimental drug; (3) have a personal history of cerebrovascular disease, brain trauma, significant endocrine disorders, cancer and/or depressive disorder; (4) have a current drug abuse or dependency disorder or in the three months prior to recruitment; and (5) have significant clinical abnormalities in routine laboratory tests.

For the French-Canadian FXS cohort, the inclusion criteria comprised (1) being male aged from 12 to 50 years old; (2) having a confirmed diagnosis of FXS; and (3) the parent/caregiver being available for the clinic visit and cognitive assessments. For the French-Canadian controls, only the first two criteria were considered. We excluded individuals treated with (1) bleeding diathesis; (2) the use of drugs that affect cholesterol metabolism such as hypolipidemic agents, immunosuppressors, retinoids, and corticosteroids; (3) malignancy; (4) liver or renal dysfunction; (5) uncontrolled hypothyroidism or hyperthyroidism; (6) malabsorption/malnutrition; (7) any acute condition or abnormalities in routine laboratory tests; and (8) any neurological or psychological disorders.

This study was approved by both the Scientific and Ethical Board of the Research Center of Centre Hospitalier Universitaire de Sherbrooke (CHUS) and the Ethical Board of the Neurology Department of FLENI hospital. Written informed consent was obtained from the caregiver/guardian on behalf of the minors and adults with FXS as well as from each participant of the control group.

### 4.2. Sample Collection

Blood samples were collected in K2EDTA tubes (BD Vacutainer^®^) for both cohorts. Plasma from the French-Canadian cohort was recovered by centrifugation of the whole blood and then stored at −80 °C until use. In parallel, plasma from the Argentinian cohort was recovered by centrifugation and kept at −80 °C. Plasma samples from Argentina were transported to Canada on dry ice and then transferred to −80 °C until FA analysis.

### 4.3. FA Extraction and Analysis

For FA quantification, a mixture of the internal standards including 25% of triglycerides (19:0), 25% of cholesteryl esters (17:0), 45.5% of phospholipids (15:0), and 4.5% of the free FAs (24:0) was added to a 200 mL plasma sample. Lipids were extracted using the Folch method [57], evaporated under N2, and then reconstituted in 200 mL chloroform. A solid-phase extraction method was used to separate and extract phospholipids, as previously described [58]. Then, phospholipids were methylated by 12% boron trifluoride methanol solution at 90 °C for 30 min to generate fatty acid methyl esters required for their analysis by gas chromatography (model 6890; Agilent, Palo Alto, CA, USA). Fatty acid methyl esters were separated on a BPX-70 fused capillary column (50 m, SGE, Melbourne, Australia) and detected by a flame ionization detector with the same parameter settings described by Chevalier et al. [59]. The chromatogram analysis was performed using the OpenLab CDS ChemStation. The results were calculated in the absolute concentration of fatty acid methyl esters compared to the internal standard (GLC-569B, Nu-Check Prep, Inc., Elysian, MN, USA).

### 4.4. Statistical Analysis

Statistical differences between the two groups for age and sex were tested using a *t*-test or Fisher’s exact test, respectively. Moreover, an age-related analysis was performed on three different age groups of children, adolescents, and adults (presented in the Appendix A). The FA data were not normally distributed; therefore, a non-parametric Mann–Whitney test was used for comparison between the two groups of FXS and controls. Multiple testing correction was performed using the false discovery rate (FDR) [60]. Statistical significance was established at an alpha level of rejection of 0.05. All statistical analyses and graphics were performed in GraphPad Prism (GraphPad Software, Version 9.3.1, San Diego, CA, USA).

## 5. Conclusions

In conclusion, the consistent findings in both Argentinian and Canadian cohorts suggest a perturbation of FA profiles in FXS individuals, in particular, for various FAs belonging to the ω-3 family. These observations are somehow consistent with previous studies on other neurodevelopmental disorders, including ASD or ADHD. Our results provide new directions for preclinical research to explore the underlying causes and effects of low plasma lipids in FXS and further explore treatment strategies, including supplementation with EPA and/or DHA.

## Figures and Tables

**Figure 1 ijms-23-10815-f001:**
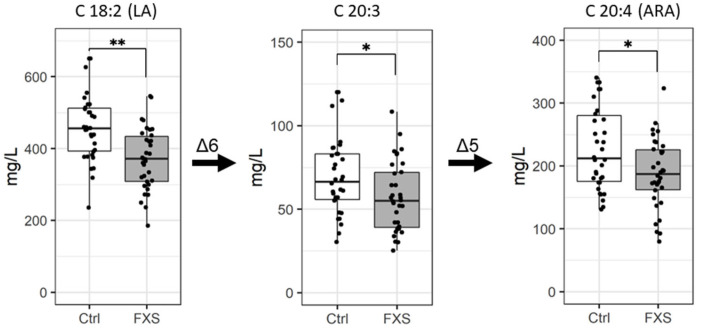
Concentrations of selected ω-6 FAs in plasma. Levels are expressed in mg/L of plasma. * *p* ≤ 0.05; ** *p* ≤ 0.01. Δ6, delta6-desaturase; Δ5, delta5-desaturase.

**Figure 2 ijms-23-10815-f002:**
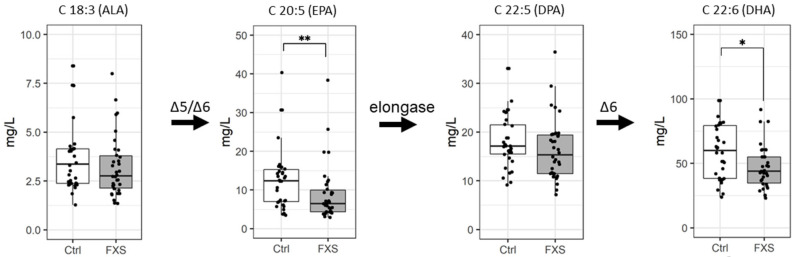
Concentrations of selected ω-3 FAs in plasma. Levels are expressed as mg/L of plasma. * *p* ≤ 0.05; ** *p* ≤ 0.01, Δ6, delta6-desaturase; Δ5, delta5-desaturase.

**Table 1 ijms-23-10815-t001:** Characteristics of Participants.

	FXS	Controls	*p*-Value
All cohorts (N)	34	34	NA
Argentinians (N)	23	23	
French-Canadians (N)	11	11	
Sex N (% males)	28 (91%)	28 (91%)	>0.99 ^1^
Age (mean ± SD)	22.8 ± 12.8	23.7 ± 13.6	0.77 ^2^
Children, N (mean yrs ± SD)	9 (6.3 ± 0.9)	9 (6.0 ± 1.2)	0.51
Adolescents N (mean yrs ± SD)	4 (14.8 ± 3.8)	4 (14.8 ± 3.8)	>0.99
Adults, N (mean yrs ± SD)	21 (31.6 ± 7.6)	21 (33.1 ± 7.1)	0.52
FXS diagnosis			
Full mutation (%)	88.6		
Mosaic (%)	11.4		

^1^ Fisher test. ^2^
*t*-test.

**Table 2 ijms-23-10815-t002:** FAs Profile in Plasma.

Fatty Acids	FXS (*n* = 34) Median in mg/L [IQ1, IQ3]	Controls (*n* = 34) Median in mg/L [IQ1, IQ3]	Adjusted *p*-Value ***
C14:0	8.3 [6.8, 9.2]	9.5 [7.4, 11.0]	0.0455
C16:0	490.8 [444.4, 530.5]	587.5 [534.2, 681.9]	0.0008
C18:0	362.7 [307.4, 409.2]	387.9 [336.4, 460.0]	0.1174
SFA total	855.1 [737.7, 962.4]	978.6 [898.4, 1070.0]	0.0017
C16:1 ω-7	6.6 [5.6, 8.4]	10.0 [8.3, 13.4]	0.0008
C18:1 ω-9	208.3 [181.9, 242.6]	242.3 [216.2, 256.0]	0.0158
C18:1 ω-7	20.7 [16.7, 26.1]	26.5 [20.3, 30.5]	0.0084
MUFA total	236.0 [202.0, 276.7]	274.4 [254.0, 297.7]	0.0084
C18:2 ω-6	372.3 [307.6, 436.7]	456.5 [389.3, 515.7]	0.0008
C20:3 ω-6	55.3 [38.9, 76.7]	66.5 [55.3, 83.9]	0.0460
C20:4 ω-6	184.6 [161.2, 226.7]	212 [173.6, 283.9]	0.0489
ω-6 total	621.8 [524.5, 717.0]	746.5 [655.2, 867.2]	0.0013
C18:3 ω-3	2.8 [2.1, 3.8]	3.4 [2.4, 4.2]	0.1844
C20:5 ω-3	6.4 [4.3, 10.2]	12.4 [6.9, 15.5]	0.0084
C22:5 ω-3	15.5 [11.5, 19,5]	17.1 [15.2, 21,8]	0.1844
C22:6 ω-3	43.8 [34.2, 55.1]	60.0 [38.0, 79.9]	0.0336
ω-3 total	72.1 [57.0, 85.0]	93.7 [68.8, 122.5]	0.0173
PUFA total	684.1 [600.0, 803.3]	842.4 [734.9, 962.3]	0.0015
ω-3/ω-6	0.1 [0.1, 0.1]	0.1 [0.1, 0.1]	0.4400
C20:5/C18:3(EPA/ALA)	2.4 [1.6, 3.6]	3.1 [2.6, 4.1]	0.0410
C22:5/C20:5(DPA/EPA)	2.5 [1.4, 3.2]	1.6 [1.2, 2.2]	0.0455
C22:6/C22:5(DHA/DPA)	2.8 [2.3, 3.3]	3.5 [2.4, 4.3]	0.1844
C22:6/C20:5(DHA/EPA)	6.6 [3.9, 10.0]	5.3 [2.7, 8.0]	0.2148
20:3/18:2	0.2 [0.1, 0.2]	0.2 [0.1, 0.2]	0.7947
20:4/20:3	3.7 [2.6, 4.5]	3.3 [2.7, 3.9]	0.6797

IQ1: interquartile range; DPA: decosapentonic acid. * Mann–Whitney test adjusted via the FDR test.

## Data Availability

Not applicable.

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
