# Peer review of "Alteration of Fatty Acid Profile in Fragile X Syndrome"

_ijms, 2022, doi:10.3390/ijms231810815_

Round 1
Reviewer 1 Report
This study aimed to assess the FAs content in plasma phospholipids in humans with FXS as compared to age and sex matched controls. The main finding is they found a decreasing trend of all measured FAs in FXS as compared to controls. I have several main concerns, as attached below.
1. The sample size is too small to draw a promissing conclusion. The multi-site data may further bias the finding.
2. It is well known that phenotype of FXS males is different to that of affected females. Also, participant age, diet habit, weight could have significant impacts on these measurements. This study should have performed more detailed conparison and assessment.
3. Though mentioned in the discussion section, no behavior assessment further decrease the significance of this study.
Author Response
Point 1. The sample size is too small to draw a promissing conclusion. The multi-site data may further
bias the finding.
Response 1. To our knowledge this is a very first report of FA profile in FXS. We agree that the
sample size in our study may not be considerable to draw conclusions on FA profile alterations.
However, FXS is a rare disease, and our sample size was comparable to previous reports on lipid
profile of FXS individuals [1]. Considering the exploratory nature of our study, we thus analyzed
the data including two cohorts from the two sites, Canada and Argentina. Since results obtained
from both cohorts showed consistent trend as the whole cohort, we agree that a larger study is
warranted to validate our findings. The small sample size limitation has been added as a limitation
in the discussion section (lines 315-317).
Point 2. It is well known that phenotype of FXS males is different to that of affected females. Also,
participant age, diet habit, weight could have significant impacts on these measurements. This study
should have performed more detailed comparison and assessment.
Response 2. We agree that sex, diet, BMI and many other factors affect FAs profile and are
important to consider. However, not all these data were available when the samples were analyzed.
In the current study we tried to diminish the effect of diet on FAs by measuring them in plasma
phospholipids which reflect the diet for last few weeks (lines 301-303). Even though the diet effect
was not controlled, having the same findings on two different populations (with different diets), we
think that diet did not account for the differences of the FAs profile between FXS individuals and
controls.
Moreover, since FXS is more prevalent in males, we had a very limited number of females which
did not allowed us to draw sex-related conclusions. We also added as a limitation the lack of BMI
data along with the effect of diet and sex in the discussion section (lines 296- 298).
Point 3. Though mentioned in the discussion section, no behavior assessment further decreases the
significance of this study.
Response 3. Thank you for pointing out the behaviour assessment. We recognize the lack of
behaviour assessment as a limitation for this study (line 310) and we suggested their inclusion for
future studies to explore the clinical significance of FAs profile alterations.
Reviewer 2 Report
In this manuscript, the authors investigate the metabolism of fatty acids in two distinct fragile X syndrome cohorts: one from Argentina and one from Canada. The rationale for performing the study is based on work in other neuropsychiatric disorders including ASD. Overall, the study design is simple and well-organized. The authors provide a very thorough explanation of the different types of fatty acids in the introduction that is very effective for laying out the study design. My few questions and concerns are listed below:
The French-Canadian cohort seems to have a more severe profile change than the Argentinians. Is there a known reason for this, e.g., any familial metabolic syndromes?
Is there still significance if females are removed from the analysis?
In Table 2, simply adding the different SFAs or other groups seems incorrect as they are not all on the same scale and certain FAs will have a greater influence on the overall sum. Instead, creating a z-score for each and then adding them up seems like a better idea.
I do have concerns about the number of 0.04 p-values that are considered “significant” particularly when the supplementary information shows that these are not consistently showing up as significant across samples.
Smaller issues:
Page 2, Line 48 – ubiquitarian should probably be ubiquitous
Page 2, Lines 84-86 – should be omitted
Page 3, Line 105 – p-value listed at 0.0014 but table says 0.0015
Page 6, Line 184 – may not want to use “mentally retarded”
Reference 37 is missing information
I think information in the header of Table 2 is cut off after “adjusted p-“.
Author Response
Response to Reviewer 2 Comments
Point 1. The French-Canadian cohort seems to have a more severe profile change than the Argentinians. Is there a known reason for this, e.g., any familial metabolic syndromes?
Response 1. Thank you for the comment. Previous studies suggest that ethnicity may influence the plasma phospholipid FAs profile, mainly the PUFA levels [1]. Differences in fish consumption between two cohorts might account for PUFAs profile differences. However, the effect of diet was not monitored in this study and was recognized as a limitation in the discussion section (lines 298-303).
Moreover, French-Canadians have a high prevalence of genetic hyperlipidemias which are not necessarily related to metabolic syndromes (line 169-176). The BMI data were not available when the samples were analyzed, and we added as a limitation in the discussion section (line 296). We suggest considering all factors that impact plasma FA profile such as diet, BMI and ethnicity for future studies in FXS.
Point 2. Is there still significance if females are removed from the analysis?
Response 2. Thank you for the question. We did not observe differences in our results when females were removed from the analysis (we added the additional results in supplementary table S1- Lines 109-110). Since FXS is more prevalent in males, we had a very limited number of females which did not allowed us to draw sex-related conclusions. This limitation was added in the discussion (lines 297-299).
Point 3. In Table 2, simply adding the different SFAs or other groups seems incorrect as they are not all on the same scale and certain FAs will have a greater influence on the overall sum. Instead, creating a - z score for each and then adding them up seems like a better idea.
Response 3. We agree that there are different scales between various FAs. However, we do not have a reference population to compute the z score and considering the small sample size, we think it would not be reliable to use the controls as the reference population.
Point 4. I do have concerns about the number of 0.04 p-values that are considered “significant” particularly when the supplementary information shows that these are not consistently showing up as significant across samples.
Response 4. We understand your concern. We recognize that the small sample size is a limitation for the study. However, we have to consider that FXS is a rare disease, and our sample size is comparable to previous reports on lipid profile of FXS individuals [2–4]. Considering the exploratory nature of the study we performed the same analysis in different sub-groups even though their sample size was too small. The latter can explain the difference of the degree of statistical significance. Although we observed the same decreasing trend in subgroups of the current study, further studies with larger sample size are warranted to further explore the alteration of FA profile.
References:
- Anderson, J.S.; Nettleton, J.A.; Herrington, D.M.; Johnson, W.C.; Tsai, M.Y.; Siscovick, D. Relation of Omega-3 Fatty Acid and Dietary Fish Intake with Brachial Artery Flow-Mediated Vasodilation in the Multi-Ethnic Study of Atherosclerosis. The American Journal of Clinical Nutrition 2010, 92, 1204–1213, doi:10.3945/ajcn.2010.29494.
- Çaku, A.; Seidah, N.G.; Lortie, A.; Gagné, N.; Perron, P.; Dubé, J.; Corbin, F. New Insights of Altered Lipid Profile in Fragile X Syndrome. PLOS ONE 2017, 12, e0174301, doi:10.1371/journal.pone.0174301.
- Toupin, A.; Benachenhou, S.; Abolghasemi, A.; Laroui, A.; Galarneau, L.; Fülöp, T.; Corbin, F.; Çaku, A. Association of Lipid Rafts Cholesterol with Clinical Profile in Fragile X Syndrome. Sci Rep 2022, 12, 2936–2936, doi:10.1038/s41598-022-07064-z.
- Lisik, M.Z.; Gutmajster, E.; SieroÅ„, A.L. Low Levels of HDL in Fragile X Syndrome Patients. Lipids 2016, 51, 189–192, doi:10.1007/s11745-015-4109-6.
Please see the attachment for the comments regarding changes in the paper.

Round 2
Reviewer 1 Report
I'd like to thank the authors for their prompt response. However, the revision doesn't sufficiently address my concerns. I won't endorse the publication of this manuscript in the current format.
Author Response
Dear Reviewer,
Thank you for the time you provided to read our responses. We respect your opinion.
We cannot make any changes in the study design currently. As we mentioned previously in responses to your comments as well as the limitation part of the project in the paper, it was not possible for us to have all those data gathered in the current study. Although we are going to have more investigations on FA profile of FXS individuals with more comprehensive data and bigger sample size in our next studies to understand the effect of diet, age, sex, metabolic disorders, etc. and verifying the results we obtained in the current study.
We hope you consider the difficulty of clinical research on rare diseases like FXS.
We value the insights and opinions you provided and appreciate it.
Best regards,
Armita
Round 3
Reviewer 1 Report
None